# A Climbing (Bouldering) Intervention to Increase the Psychological Well-Being of Adolescents in the Bekaa Valley in Lebanon-Study Protocol for a Controlled Trial

**DOI:** 10.3390/ijerph20054289

**Published:** 2023-02-28

**Authors:** Katharina Luttenberger, Charbel Najem, Simon Rosenbaum, Charles Sifri, Leona Kind, Beat Baggenstos

**Affiliations:** 1Department Medical Psychology and Medical Sociology, Faculty of Medicine, Friedrich-Alexander-Universität Erlangen-Nürnberg, 91054 Erlangen, Germany; 2Spine, Head and Pain Research Unit Ghent, Department of Rehabilitation Sciences, Faculty of Medicine and Health Sciences, Ghent University, 9052 Ghent, Belgium; 3Department of Physiotherapy, Faculty of Public Health, Antonine University, Baabda, Lebanon; 4Discipline of Psychiatry and Mental Health, Faculty of Medicine and Health, UNSW, Sydney 2052, Australia; 5ClimbAID Lebanon, Branch of ClimbAID, 8048 Zurich, Switzerland

**Keywords:** adolescent mental health, physical activity, bouldering/climbing, refugees, intervention studies

## Abstract

(1) Background: Adolescent refugees in Lebanon and Lebanese youth are both at high risk of suffering from reduced psychological well-being. Sport is an evidence-based strategy for improving mental and physical health, and climbing is a type of sport that may positively impact both. The aim of this study is to test the effect of a manualized, psychosocial group climbing intervention on the well-being, distress, self-efficacy, and social cohesion of adolescents in Lebanon. In addition, the mechanisms behind psychological changes will be investigated. (2) Methods: In this mixed-methods waitlist-controlled study, we are allocating a minimum of 160 participants to an intervention (IG) or a control group (CG). The primary outcome is overall mental well-being (WEMWBS) after the 8-week intervention. Secondary outcomes include distress symptoms (K-6 Distress Scale), self-efficacy (General Self-Efficacy Scale; GSE), and social cohesion. Potential mechanisms of change and implementation factors are being investigated through qualitative interviews with a subgroup of 40 IG participants. (3) Conclusions: The results may contribute to knowledge of sports interventions and their effects on psychological well-being and will provide insights regarding low-intensity interventions for supporting adolescent refugees and host populations in conflict-affected settings. The study was prospectively registered at the ISRCTN platform (current-controlled trials). ISRCTN13005983.

## 1. Introduction

Lebanon is severely burdened by an ongoing economic crisis, political gridlock, hyperinflation, unemployment, and an outdated infrastructure. Problems have been significantly amplified by the 2020 port explosion and the COVID-19 pandemic. In an attempt to protect an already underfinanced healthcare system, the Lebanese government enforced a strict lockdown that may have saved lives from COVID-19, but led to an exacerbated crisis in addition to the already struggling economy, and interrupted education for approximately 1.5 million children [1]. The Russian invasion of Ukraine in 2022 exacerbated the situation through a wheat crisis and an acute shortage of bread. People were forced to stand in line for several hours to obtain sufficient staples to feed a family [2]. At the same time, Lebanon has taken in the more refugees in relation to its population size than other country in the world [2]. Among these refugees are an estimated 1.5 million displaced Syrians and another 0.5 million Palestinians. More than 50% of the refugee population in Lebanon are children [3].

In the Bekaa region of Lebanon, where most Syrian refugees reside, the situation for both the Lebanese host community and Syrian refugees has deteriorated throughout the ongoing economic crisis. Nearly 70% of income-generating household members in the Bekaa are children between the ages of 4 and 18 years, and therefore, around 40% of Syrian refugee children are not enrolled in any type of schooling [1]. Concurrently, 77% of Lebanese and 99% of Syrian households in Lebanon reported not having sufficient food or money in March 2021 [4]. Due to the worsening economic and political situation, there is a deterioration of the relationship between the Lebanese host and the Syrian refugee communities [5].

Adverse living conditions, human rights violations, and ongoing stressors have a detrimental effect on mental health. Children and adolescents are an especially vulnerable group [6]. Considering that mental and physical symptoms are interrelated [7], and that poor mental health is associated with a reduction in life expectancy [8], protecting and promoting the mental health of adolescents, especially in resource-poor settings is of high importance. Services offered by nonprofit organizations and national health service providers typically have a limited reach, leaving many young refugees without access to support programs [9]. There is a need for scalable, low-threshold, evidence-based psychosocial interventions that simultaneously strengthen mental health in addition to physical health [10].

In nonhumanitarian settings, the positive effects of sport and physical activity have been widely researched. Regular participation in physical activity and sport is associated with improvements in symptoms of depression in adults [11] and adolescents [12], post-traumatic stress symptoms [13], anxiety disorders [14], eating disorders [15,16], and substance dependence [17]. In addition, physical activity can protect against the onset of mental illness [18,19]. Especially for disadvantaged youth, sport is a protective factor [20]. Physical activity is further capable of improving self-esteem, energy levels, well-being, and overall quality of life [21,22,23]. As a psychosocial intervention, sports programs create a healthy environment where participants can manage their stress in a positive way, form team friendships, and experience a sense of belonging. Moreover, regular physical activity can improve self-efficacy, emotional intelligence [24,25], and it is positively correlated with social cohesion [26]. Regular physical activity is associated with psychological well-being and reduced levels of psychological distress [27,28]. The existing literature illustrates that sport and physical activity in the context of humanitarian assistance are capable of improving the psychosocial well-being of young refugees [29]. For example, a Capoeira dance/martial arts program for young refugees in Australia had positive effects on interpersonal skills, sense of responsibility, and discipline [30]. Furthermore, a surf therapy program showed improvement in children’s well-being [31]. Specific cultural beliefs about physical activity, which often arise from different ethnic identities and exposures, play a strong role in people’s decision to adopt a physical activity program [32,33,34,35]. For survivors of torture, trauma-sensitive soccer proved to be a critical factor for regaining access to one’s body, building social relationships, and learning to cope with stressful events [36]. Moreover, a study involving children and adolescents found that after disasters, sports and exercise are able to promote emotional and social stabilization and improve resilience [37]. However, all these previous studies relied on uncontrolled pre-post methodologies. Sport is also a highly accepted and culturally appropriate way to reduce stress, yet access to sport is often severely limited, particularly in camp environments [9,23], which is a risk factor for poor mental health in and of itself [38]. It has also been shown that opportunities to participate in sports are usually limited to specific refugee groups such as able-bodied young men, which limits participation of other groups in particular, children, and females [39]. Therefore, targeted programs are required. Unfortunately, existing research largely focuses on refugees living in high-income countries, despite low- and middle-income countries (LMICs), such as Lebanon, hosting the greatest number of refugee and the associated need for health promotion initiatives [28].

Climbing as therapy has been shown to reduce symptoms of depression in adults in at least two studies involving randomized controlled trials [40,41] and has shown a tendency to improve mental health in general in a recent meta-analysis [42]. It also has the potential to enhance feelings of self-efficacy [43], which is often limited in disadvantaged youth. This paper presents the design of a mixed-methods, waitlist-controlled intervention study on psychological well-being, distress, and self-efficacy, as well as social cohesion in underprivileged adolescents (refugees and the host community) living in Lebanon.

## 2. Methods

### 2.1. Overall Aims

The aim of the study is to test the effectiveness of a climbing intervention among adolescent locals and refugees living in the Bekaa region of Lebanon. We hypothesized that the climbing intervention will have a beneficial effect on participants’ psychological well-being. Additionally, we are investigating the impact of the intervention on, and the relationships between, self-efficacy, psychological distress, participants’ perceptions of other nationalities (Syrian, Lebanese, Palestinian), and social cohesion.

To gain insight into the modes of action and potential benefits that are specific to culture and living conditions as well as barriers and facilitators of the intervention, we are conducting qualitative interviews with a subsample of participants, balanced for age, sex, and intervention subgroup.

### 2.2. Trial Outcomes and Hypothesis

#### 2.2.1. Primary Outcome

Psychological well-being is being measured with the WEMWBS [44]. The participants in the intervention group (IG) are expected to benefit significantly more in terms of their psychological well-being than adolescents assigned to the control group (CG; effect size of Cohen’s d ≥ 0.55).

#### 2.2.2. Secondary Outcomes

Symptom severity of distress, self-efficacy, and social cohesion outcomes are being measured. Additionally, qualitative interviews are being conducted with at least 30 members of the IG and analyzed using content analysis as described by Mayring [45].

#### 2.2.3. Study Design and Setting

The research is designed as a waitlist-controlled trial, with the implementation of a manualized climbing intervention for Lebanese adolescents and adolescent refugees living in the Bekaa region of Lebanon. The plan is for the CG not to receive any study-specific intervention while the IG receives the intervention, but the CG members will be invited to participate in an upcoming cycle of the climbing program. Members of the CG will, therefore, be offered the same intervention as the IG after their waiting period. For the individual participant, the study duration will be up to approximately 12 weeks with 8 weeks of intervention (see Figure 1), followed by the voluntary participation of the CG members in the intervention. The procedure involves administering surveys to both groups (IG and CG) before the intervention begins (t0, pretest) and immediately after the intervention ends (t1, posttest). During the second half of the intervention period, qualitative semi-structured interviews will be conducted with members of the IG focusing on modes of action and outcomes of the intervention. The participant timeline is presented in Figure 1. Trial Registration Data are presented in Table 1 (ISRCTN13005983, registered April 2022, registered prospectively).

### 2.3. Eligibility Criteria

The target population of the study includes adolescents (refugees and those from the host community) living in the Bekaa valley in Lebanon.

Inclusion criteria:Between the ages of 14 and 19;Informed consent for participation in the study from adolescents as well as legal guardians (adult relatives), particularly for data collection, pseudonymized data storage, and analysis;Availability and the ability to come to the climbing intervention and to participate in the data collection;Living in the area limited by Zahle (N), Deir Zenoun, Marj (S), and Qab Elias/Bouarej (W).

Exclusion Criteria:Physical contraindications (pregnancy or other medical conditions that preclude climbing);BMI < 18.5 or >35.

### 2.4. Screening and Enrollment

To provide access to the climbing program to as many adolescents as possible and to ensure high representativeness of the sample, the adolescents are being recruited through several outreach channels (e.g., schools, other non-profits, social networks of current participants, social media, outreach campaigns in informal tented settlements). Informational material is being distributed and events are being held in schools and refugee camps to inform adolescents about the climbing program. A social media campaign is also being launched. Parents or (legal) guardians can use an online form to apply to participate in the program and, thereby, provide informed consent. If the inclusion criteria are met, the adolescents are then invited to attend a first trial session to determine whether they want to participate. If they are interested in participating, inclusion and exclusion criteria are verified by the study personnel, and the adolescents and their parents are asked to provide written informed consent. The written informed consent of the parents/legal guardians are gathered through personal communication between a trained staff member of ClimbAID and the respective guardian to make sure that every participant and their legal guardians understand the study conditions.

### 2.5. Intervention

Since 2017, the “Climbing for Peace” project of the Swiss nonprofit organization ClimbAID has been using climbing as a tool to build inclusive communities, improve mental well-being, and address social problems in projects with young people from host and refugee communities in Lebanon. The PSS program “YouCLIMB” investigated in this study is based on climbing therapy and experiential education. It was developed to improve the physical and mental well-being of young people and to help them develop social and life skills. The activities are being led by four local facilitators who are trained by a certified climbing instructor (Swiss J + S Course Instructor Sport Climbing). In addition to the sport-specific training, the coaches are trained in PSS facilitation, child protection, and mental health first aid and are supervised by a social worker and a climbing therapist.

Each session begins with a welcome, a reminder of the group rules, and an introduction to the theme and goals of the session. After a breathing meditation exercise, there is a warm-up routine, which is jointly led by the group. In the activity part of the session, the group works through experiential climbing activities and games that are designed to convey and experience the objectives of the session. 

Session 3, for example, features the game “Three-legged Climbing” to demonstrate the module topic of teamwork and cooperation. In this game, two players are connected by a string at their ankles and must work together to climb on a climbing wall. Success requires full focus, empathy, clear communication, trust, and respect—all elements of teamwork and cooperation.

Sessions 5 and 6 focus on communication and conflict resolution, providing theoretical insights such as the “four sides of communication” by Friedemann Schulz von Thun [46] and “compassionate communication” by Marshall Rosenberg [47]. These concepts are applied through activities and games on and off the climbing wall, followed by group discussions to reflect on experiences and apply them to real-life situations. After a mindfulness or meditation exercise, the session ends. The associated session-specific topics can be found in Table 2.

The CG initially receives no climbing intervention, but all participants are rewarded with 100,000 LBP (between $2.80 and $3.60 USD depending on the daily rate) after completing the posttest. After completing the IG-CG climbing cycle (8 weeks), participants in the CG will also be given an opportunity to participate in the YouCLIMB program.

### 2.6. Allocation to the IG and CG

Due to culture-specific requirements and for organizational reasons, stringent randomization is not possible. For example, due to school exams, many participants are limited to only one of the three cycles. Some female participants are only allowed to participate if they are accompanied by another member of the family. To navigate this and to increase adherence to the intervention, the timing of the intervention (immediately = IG, 3 months later = CG) can be self-selected by participants, providing they are available at each time point when data are being collected. In addition, we are accommodating special requirements (e.g., women-only groups, groups for those who can only participate if accompanied by a specific person). Out of the actual eligible applicants, about 20% stated that their availability was limited to one of the three cycles. We are allowing such applicants to self-select the timing of their participation, thereby determining their classification as either CG or IG participants.

Remaining participants are being allocated randomly. First, each applicant is being assigned a random number by using Microsoft Excel’s “=RAND()” function. Then, applicants are being allocated to the CG or IG based on the parity of the random number’s thousandth digit, whereby if the thousandth digit is even, the applicant is allocated to the CG, otherwise the participant is allocated to the IG. 

### 2.7. Sample Size Estimation

Power calculation was performed using G*Power software [48] (University of Duesseldorf, Germany) with a two-tailed alpha of 0.05, a beta of 90%, and an estimated effect size of Cohen’s d = 0.55. This resulted in a necessary sample (as treated AT) of at least 52 members per group at posttest. 

As the dropout rate is difficult to foresee in the current political and economic situation, we assumed a drop-out rate of 35% per intervention cycle. Thus, we need to enroll 80 participants in the IG and the CG, respectively. Hence, at least 160 adolescents need to be included in the study, distributed across the six climbing groups (each consisting of 12–14 participants) and the control group. Including the waitlist-control group (which will be offered the intervention after the study period), up to a total of 160 adolescents should thus be provided with a climbing program. Figure 2 gives an estimated overview about sample sizes.

### 2.8. Data Collection

Data collection is being conducted in groups of up to 12 participants with trained study personnel. Paper questionnaires are given to the participants. All literate participants are asked to fill out the questionnaires on their own. For those with reading difficulties, the questions are read aloud, and the response options are explained. The study personnel are trained not to influence the answers. All participants sit as far from each other as is necessary to safeguard their anonymity. All participants are informed that no one (i.e., including their parents, friends, or any government or nongovernment organization) apart from ClimbAID will be given information about their answers. The trained study personnel have the same cultural background (Syrian or Lebanese) as the participants to reduce misunderstandings due to different Arabic dialects. Complete data collection is also planned for dropouts, if available.

### 2.9. Measures

#### 2.9.1. Primary Outcome Measure

Overall mental well-being is measured with the Arabic version of the WEMWBS (Warwick-Edinburgh Mental Well-Being Scale) at both time points of the study. The WEMWBS consists of 14 items covering the emotional and functional aspects of mental well-being. Sum scores range from 14 to 70 with higher scores indicating higher overall mental well-being. An improvement of 3 to 8 points is regarded as clinically relevant [44]. 

#### 2.9.2. Secondary Outcome Measure

Psychological distress is measured with the Arabic version of the K-6 Distress Scale [49,50]. The K-6 consists of six items covering the main depressive symptoms (i.e., hopelessness, feeling worthless, nervous, so sad that nothing can cheer me up, and the feeling that everything is an effort). Sum scores range from 6 to 30 with higher scores signifying higher psychological distress. The authors of the Arabic validation study used 16.25 as the cut-off score for psychological disorders.

Self-efficacy is measured with the Arabic version of the general self-efficacy scale (GSE) [51]. The GSE measures optimistic self-beliefs about coping with a variety of difficult demands in life. Its 10 items can be rated on a 4-point scale ranging from 1 (not at all true) to four (exactly true). The total score ranges from 10 to 40 with higher scores indicating higher self-efficacy. The GSE has good psychometric properties [52].

Social cohesion is measured with a five-item set and an additional single-item measure from the ARK Regular Perception Survey. This survey is used specifically to analyze social tension throughout Lebanon. The items were modified to fit the situation experienced by adolescents. The five-item set uses a 5-point Likert scale ranging from 1 (very disagreeable) to 5 (very agreeable) to ask how agreeable five different scenarios are. The questions always refer to members of the other nationality, and the scenarios involve playing with or attending school with children of the other nationality, living next door to or sharing a workplace with a member of the other nationality, or having a family member who marries someone of the other nationality. In the additional single-item measure, participants are asked to rate how much they agree with the phrase: “Lebanese and Syrians share many values and have compatible lifestyles” on a 4-point Likert scale ranging from 1 (strongly agree) to 4 (strongly disagree) [5]. 

The following variables are recorded as possible variables of influence or confounding variables in the questionnaire format:Sociodemographic data: sex, age, school enrollment, nationality, living conditionsPhysical activityMedical conditionsClimbing experience

In addition, qualitative data are being gathered from at least 30 participants in the intervention groups. The sample is balanced according to age and sex, and participation is voluntary. These data are being collected via semi-structured qualitative interviews by trained personnel who are not involved in the intervention to investigate the acceptance and factors that affect the climbing program. The interview guidelines focus on the categories Reach, Effectiveness, and parts of the implementation according to the RE-AIM scheme by Holtrop et al. [53]. It is divided into main questions, possible follow-up questions, and probes. The main questions are: Reach: “Can you tell us how it came about that you are now participating in the YouCLIMB Program?” Effectiveness: “Can you tell us about your experience with the YouCLIMB program?” Implementation: “Were there any barriers that prevented you from participating or that made participating difficult?” Closing question: “Is there anything else you want to tell me?”

### 2.10. Data Quality Management

All data collectors and qualitative interviewers, as well as the facilitators are thoroughly trained for their respective tasks by the study headquarter and the ClimbAID organization. Treatment adherence is documented with detailed session protocols. 

Quantitative data and all identifiable data remain within the ClimbAID organization. Only members of ClimbAID have access to the lists of the participants’ names and codes. All analyses are conducted on pseudonymized data. Published material will not contain any patient-identifying information.

### 2.11. Data Analysis

#### 2.11.1. Quantitative Data

The data analyses will be performed with the “IBM SPSS Statistics 28” software. To assess the quality of the group allocation, the baseline data from the intervention and control groups will be examined for statistically significant differences. All data will be investigated for plausibility. Participants who drop out of the study but are still available for the posttest will be interviewed subsequently. A missing data evaluation will be carried out, and missing values will be imputed with EM-Imputation [54]. The primary data analytic strategy will be “per protocol”. As a sensitivity analysis, an additional analysis with “intention to treat” will be performed. The level of statistical significance will be set at *p* ≤ 0.05. Mixed ANOVAs (time × group) will be used to determine differences between the groups. Additionally, regression analyses will be used to test the hypotheses and check for possible confounders. Differences between groups will be determined before analysis, and if there are significant differences in important sociodemographic variables, appropriate statistical methods (e.g., propensity score matching) will be used. Secondary outcomes will be tested in an exploratory fashion with *t* tests for independent samples.

#### 2.11.2. Qualitative Data

Qualitative interviews will be analyzed with content analysis as described by Mayring [55], adhering to the guidelines of thematic analysis [56]. Therefore, a team of researchers will first deduct possible categories through an analysis of the literature and then test this set of categories with 5–10% of the interviews and develop new categories from the interviews (bottom-up). In consensus meetings, the final set of categories will be defined and tested with the next 10% of interviews to rate the amount of concordance between the different raters. Qualitative and quantitative data will be related to each other to identify categories that are related to improvement.

### 2.12. Ethical Considerations

The authors assert that all procedures contributing to this work comply with the ethical standards of the relevant national and institutional committees on human experimentation and with the Helsinki Declaration of 1975, as revised in 2008. All procedures involving human participants were approved by the responsible Ethics Committee of the Antonine University Baabda (approval number 279-2022 in February 2022). 

Informed consent is being obtained from all study participants before they enroll in the study. For minors, consent is being obtained along with written parental or caregiver consent. When the data are analyzed and the results of the study are published, the names and identities of the participants will not be used. Participants are free to leave the study at any time without supplying a reason and with no further consequences.

In case of an accident requiring medical treatment during the intervention, the participant will be taken to an appropriate medical care provider. Fees will be paid by ClimbAID and the insurance of its Lebanese partner Arcenciel (owner of the land and the facility).

Definition of adverse events (AEs): Mild AEs in the IG: all sorts of bruises or scratches or other minor superficial injuries that are transient and do not require treatment;Moderate AEs: injuries that are transient but require medical treatment, such as ligament ruptures or sprains, broken bones in legs or arms;Severe adverse events (SAEs): severe head injuries, spinal cord injuries, death, suicide attempts.

Termination rules: If an SAE occurs in the IG during the intervention, the program will be terminated if the intervention is determined to be directly responsible for the SAE.

In the case of moderate AEs, the intervention will be terminated if there are three more AEs in the intervention than in the waitlist group, if the AEs are determined to be a direct result of the intervention. Mild AEs will be recorded but will not result in the termination of the program.

## 3. Discussion

This is the first study to investigate the effect of a psychosocial bouldering intervention with adolescent refugees. The intervention is being carried out in Lebanon, which is one of the countries with the highest per capita intake of refugees [2] and will therefore contribute to knowledge about the situation of refugees in low/middle income countries [38]. With a waitlist-control-group design, all participants will be able to benefit from the opportunity at no cost. The existing literature shows that many Syrian refugees do not meet the WHO’s physical activity guidelines [57]. About 55% of Syrian refugee adolescents [2] and at least 30% of Lebanese youth [58] experience at least one psychiatric diagnosis, such as depressive symptoms, anxiety, PTSD, or related symptoms (e.g., sleeping disorders). In addition to the psychosocial intervention provided in a safe and protected environment, participants may also profit from the increased physical activity.

The results of the study may contribute to knowledge about sport interventions and their effects on psychological well-being. Knowledge regarding novel, low-threshold intervention to support adolescent refugees residing in disadvantaged surroundings is urgently needed. 

The climbing intervention has a slight risk of injury, comparable to the risk involved in other sport interventions. In previous studies conducted by our own research group with around 5000 bouldering hours, no severe adverse events have occurred. Nevertheless, any accidents and potential AEs, including suicidal behavior or accidents resulting in injuries, will be documented by the trainers/facilitators for the IG and by the study headquarters for the CG. Regular interim evaluations will be made, and in the case of SAEs in the IG, the intervention will be terminated. As with any activity involving transportation in a motor vehicle, there is always a risk of injury, permanent damage, or death from accidents. The responsible NGO always tries to keep the risk of accidents as low as possible and therefore only works with reliable and trustworthy transportation service providers who comply with the relevant legal provisions and safety regulations. Participants are brought by exclusive buses to and from the intervention, and the bus driver is accompanied by ClimbAID staff to provide a safe environment for all participants, especially female participants. In the case of an AE requiring medical treatment, ClimbAID will cover the patient’s costs, thus ensuring adequate treatment.

The strengths of the study are the large sample size, the existence of a control group, the mixed-methods approach, the use of manualized treatment options, the naturalistic setting, and therefore, the pragmatic nature of the evaluation of a real-world program. One limitation is the lack of randomization. In this specific study, randomization may also have been a limitation because it may have resulted in the exclusion of female participants for cultural reasons. Being female in a refugee context is a risk factor for mental distress, especially depression [38]. Syrian refugees in Lebanon often live according to religious or cultural traditions with a strong desire to protect girls from potential harmful encounters outside the family. To include female participants, we decided not to randomize the group to allow family members who also need to comply with the inclusion criteria (e.g., mostly brothers, sisters, cousins) to participate together in the same intervention group. In addition, we offered female-only groups, and ClimbAID staff travel on the buses that collect all the participants from their homes. No participants must use public transportation. We therefore concluded that the disadvantages of randomization (which would prevent a vulnerable group from participating in the intervention) outweighed the positives. If the groups differ at baseline in relevant factors, appropriate statistical methods (e.g., propensity score matching) will be applied.

Another potential limitation is that the bouldering intervention cannot be blinded with respect to facilitators or participants. All outcome measures are self-assessed and, therefore, are also not blinded. Third, in the absence of exclusion criteria regarding possible diagnoses or levels of psychological distress, it is possible that we will obtain a heterogeneous sample, with the associated advantages of a naturalistic design and low-threshold intervention and the disadvantages of potentially diluted intervention effects.

## 4. Conclusions

The study is expected to contribute knowledge on the implementation of a psychosocial boulder intervention and its impact on psychological well-being, stress and self-efficacy among adolescent refugees. This can also generate knowledge about the implementation of low-threshold interventions to support adolescent refugees and the host population, as well as social cohesion in conflict-affected areas.

## Figures and Tables

**Figure 1 ijerph-20-04289-f001:**
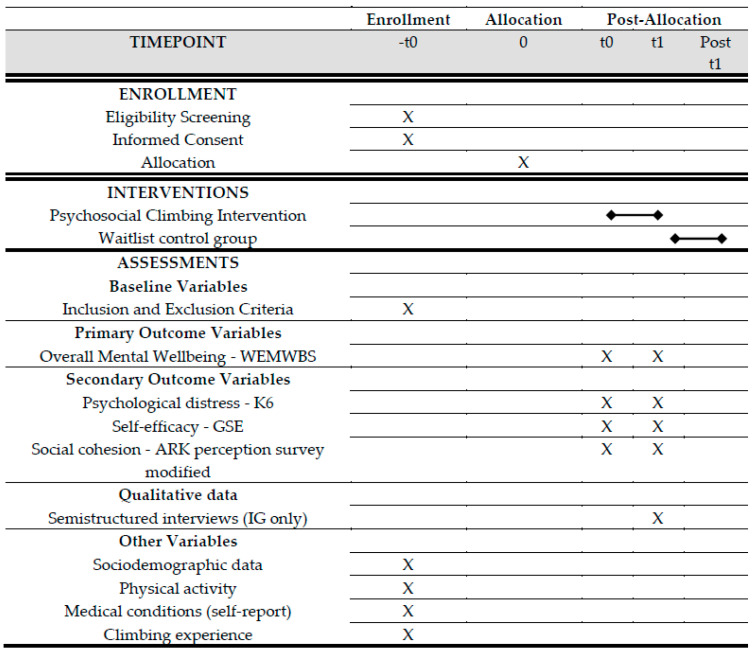
SPIRIT Participant timeline.

**Figure 2 ijerph-20-04289-f002:**
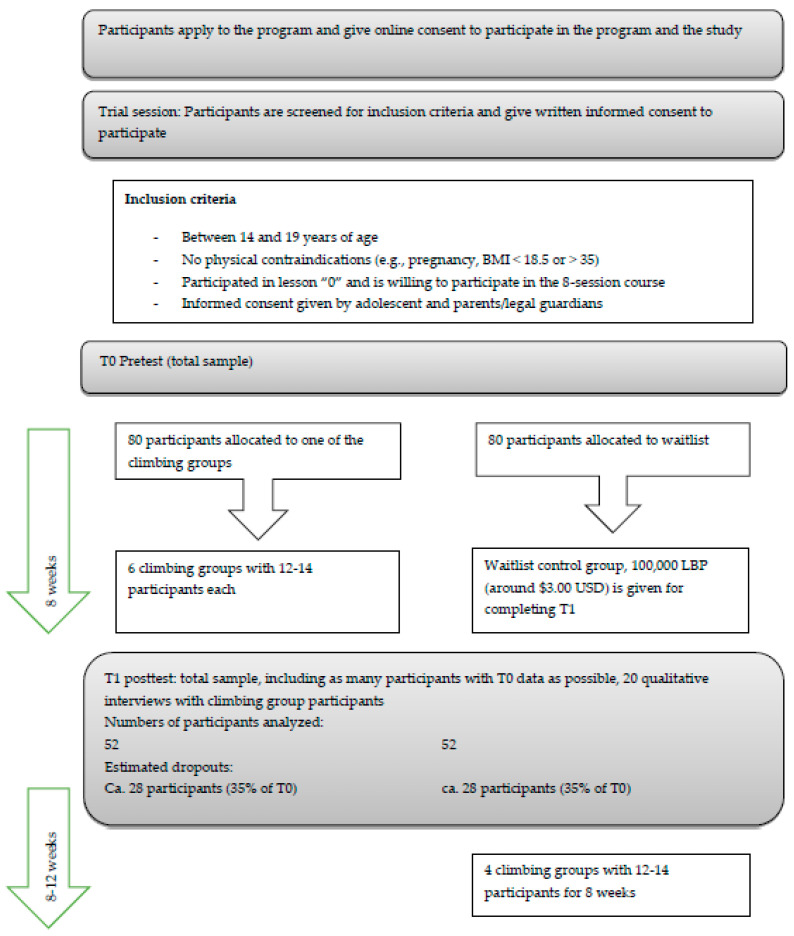
Consort Flow Chart with Estimated Numbers of Participants.

**Table 1 ijerph-20-04289-t001:** Trial Registration Data according to SPIRIT guideline.

Data Category	Information
Primary registry and trial identification number	ISRCTN13005983
2. Date of registration in primary registry	29 March 2022
3. Secondary identifying numbers	-
4. Source(s) of monetary or material support	Mammut Sports Group
5. Primary sponsor	Universitätsklinikum Erlangen, Germany
6. Secondary sponsor(s)	Antonine University, Hadat Baabda, Lebanon
7. Contact for public queries	see Point 8
8. Contact for scientific queries	PD Dr. Katharina Luttenberger, katharina.luttenberger@uk-erlangen.de
9. Public title	Effectiveness of a climbing (bouldering) intervention on psychological wellbeing for adolescents in the Bekaa Valley, Lebanon: (How) does it work?
10. Scientific title	Mixed method study on a climbing (bouldering) intervention to increase the psychological well-being of adolescents in the Bekaa Valley in Lebanon, waitlist-controlled trial.
11. Countries of recruitment	Lebanon
12. Health condition(s) or problem(s) studied	Psychological wellbeing of adolescent in Bekaa, Lebanon, refugees and host community
13. Intervention(s)	Study arm 1: Intervention group receiving the psychosocial bouldering intervention
	Study arm 2: waitlist control group receiving the intervention after posttest.
14. Key inclusion and exclusion criteria	Ages eligible for study: adults; Sexes eligible for study: both
	Inclusion criteria: 1. Aged between 14 and 19 years 2. Written informed consent of parents or young adult 3. Ability to reach the climbing intervention
	Exclusion criteria: 1. physical contradictions to climbing
15. Study type	Controlled intervention study
16. Date of first enrollment	April 2022
17. Target sample size	160
18. Recruitment status	complete
19. Primary outcome(s)	Overall mental wellbeing is measured with the WEMWBS (Warwick-Edinburgh Mental Well-Being Scale) at baseline and after the intervention (8 weeks)
20. Key secondary outcomes	Distress severity, general self-efficacy, social cohesion

**Table 2 ijerph-20-04289-t002:** Session-specific topics in the bouldering intervention.

Session	Module Themes	Cross-Cutting Themes
1	Trust & respect	Shared leadership Emotional education Mindfulness
2		
3	Cooperation & teamwork	
4		
5	Communication & conflict resolution	
6		
7	Problem-solving & decision-making	
8		

## Data Availability

The research group intends to publish data generated from this study in open-access, peer reviewed journals. The datasets which will be used and/or analyzed during the current study are available from the corresponding author on reasonable request after the publication of the results. Trial registry will be updated if protocol modifications are made. Model consent forms in Arabic language were approved by the Ethics Committee and will be made accessible upon request.

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
