# Peer review of "A Climbing (Bouldering) Intervention to Increase the Psychological Well-Being of Adolescents in the Bekaa Valley in Lebanon-Study Protocol for a Controlled Trial"

_ijerph, 2023, doi:10.3390/ijerph20054289_

Round 1

Reviewer 1 Report

The introduction should describe the links between sports activities and self-efficacy, psychological distress, perception of other nationalities and social cohesion. Some of the analyzed variables appear only in the Method part.

Line 139 – the posttest measurement must be named t1 instead of t2.

Figure 1 is actually a table.

Tables 1 (Figure 1) and 2 are hard to follow. Perhaps it would be better to replace them with the textual description of the data.

Figure 2 repeats the information already presented in the text.

The Results part is missing!

Author Response

Dear reviewer,

thank you for yur comments. Please find our answers below, marked in green.

1. The introduction should describe the links between sports activities and self-efficacy, psychological distress, perception of other nationalities and social cohesion. Some of the analyzed variables appear only in the Method part.
We included some sentences and literature (Ref 24-28) to the relation between physical activity and social cohesion (133 ff and 141 ff). Also the already existing links between the other variables are strengthened (line 124-148)

Line 139 – the posttest measurement must be named t1 instead of t2.c
Thank you! We corrected that

Figure 1 is actually a table.
SPIRIT guidelines recommend this wording and we adhere exactly to them. Please see here: Schedule of enrolment, interventions, and assessments – GUIDANCE FOR CLINICAL TRIAL PROTOCOLS (spirit-statement.org)

Tables 1 (Figure 1) and 2 are hard to follow. Perhaps it would be better to replace them with the textual description of the data.
We assume you are referring to our current Figure 1 and Table 1. See above, we follow the SPIRIT guidelines, as should be done when presenting the design of controlled intervention studies. Data set – GUIDANCE FOR CLINICAL TRIAL PROTOCOLS (spirit-statement.org)

Figure 2 repeats the information already presented in the text.
We believe that Figure 2 contributes to the understanding of the study design, but if the editor feels that Figure 2 is superfluous, we are happy to remove it.

The Results part is missing!
Correct. As this is a design paper, we are still conducting the study and have no results yet.

Reviewer 2 Report

As the protocol was successfully funded, there are no significant issues identified. 

However, as this is a physical activity (climbing) intervention, detail description of how Session-specific topics in the bouldering intervention will be input into the intervention should be written. Moreover, the quantitative analysis is not sufficient, repeated measure ANOVA should be used rather than just t-test and regression. 

Author Response

Dear Reviewer,

thank you for your comments. Please find our answers below in green.
As the protocol was successfully funded, there are no significant issues identified. 
Thank you

However, as this is a physical activity (climbing) intervention, detail description of how Session-specific topics in the bouldering intervention will be input into the intervention should be written.
We gratefully took up your suggestion and inserted a part into the “intervention” section to demonstrate how session-specific topics are learned in the bouldering session (lines 506-516)

Moreover, the quantitative analysis is not sufficient, repeated measure ANOVA should be used rather than just t-test and regression. 
We included a mixed ANOVA (time x group) instead of the t-test. Line 734